# Experimental Study on the Exploration of Camera Scanning Reflective Fourier Ptychography Technology for Far-Field Imaging

**Mingyang Yang [1,2], Xuewu Fan [1], Yuming Wang [1,2] and Hui Zhao [1,*]**

[1] Space Optical Technology Research Department, Xi'an Institute of Optics and Precision Mechanics, Chinese Academy of Sciences, 17 Xinxi Road, Xi'an 710119, China; yangmingyang@opt.cn (M.Y.); fanxuewu@opt.ac.cn (X.F.); wangyuming2016@opt.cn (Y.W.)

[2] School of Optoelectronics, University of Chinese Academy of Sciences, Beijing 100049, China

[*] Correspondence: zhaohui@opt.ac.cn; Tel.: +86-153-5354-6148

**Abstract:** Fourier ptychography imaging is a powerful phase retrieval method that can be used to realize super-resolution. In this study, we establish a mathematical model of long-distance camera scanning based on reflective Fourier ptychography imaging. In order to guarantee the effective recovery of a high-resolution image in the experiment, we analyze the influence of laser coherence in different modes and the surface properties of diverse materials for diffused targets. For the analysis, we choose a single-mode fiber laser as the illumination source and metal materials with high diffused reflectivity as the experimental targets to ensure the validity of the experimental results. Based on the above, we emulate camera scanning with a single camera attached to an X-Y translation stage, and an experimental system with a working distance of 3310 mm is used as an example to image a fifty-cent coin. We also perform speckle analysis for rough targets and calculate the average speckle size using a normalized autocorrelation function in different positions. The method of calculating the average speckle size for everyday objects provides the premise for subsequent research on image quality evaluation; meanwhile, the coherence of the light field and the targets with high reflectivity under this experiment provide an application direction for the further development of the technique, such as computer vision, surveillance and remote sensing.

**Keywords:** fourier ptychography; coherent illumination; camera-scanning; high-resolution imaging; synthetic aperture

## 1. Introduction

Optical synthetic aperture (OSA) is achieved by combining a series of easily fabricated small aperture systems stitched together into a large aperture to achieve high resolution imaging requirements. OSA can be divided into three forms of implementation according to segmented mirror, sparse aperture and phased array. At present, the world's leading scientific and technological countries have taken synthetic aperture as one of the main research subjects for realizing high-resolution (HR) imaging systems and have made many breakthroughs. In the 1970s, Meinel designed a large infrared telescope, which consisted of six independent telescopes to synthesize a 5.6 m aperture, which has successfully opened up the theoretical research of modern optical synthetic aperture technology [1]. For segmented mirror form, the Thirty Meter Telescope (TMT) uses the technique and reflective structure with a primary mirror consisting of 492 hexagonal mirrors to the equivalent aperture of 30 m, and the TMT focal length is 450 m with an effective field of view (FOV) of 20 arcmin under a wavelength of 0.31–28 μm [2]. Among the space-based systems, the James Webb Telescope (JWST) also uses segmented mirror technology, the primary mirror of which consists of 18 hexagonal mirrors with a diagonal distance of 1.5 m and the equivalent aperture of 6.5 m [3]. For sparse aperture form, among ground-based systems, the Giant Magellan Telescope (GMT) is designed using the sparse aperture optical system; the primary mirror consists of seven mirrors of 8.4 m aperture with an equivalent aperture

of 24.5 m, and the secondary mirror consists of seven concave ellipsoidal mirrors with a diameter of 1.1 m and an equivalent aperture of 3.2 m [4]. For phased array form, a multi aperture imaging array (MAIA) system is combined with nine sub-telescopes of 10 cm aperture arranged in a "Y" shape. The Star9 system is also composed of nine sub-telescopes, each of which has an aperture of 12.5 m and a field of view of 150 mrad. Each of sub-telescopes has an aperture of 12.5 cm with a system equivalent aperture of 0.61 m and a field of view of 1 μrad [5]. There is also the example of synthetic aperture using phased array, which is conducted by NASA on space-based telescope array systems and was used to design the space interferometer mission (SIM), which is made up by three parts, a science interferometer (SI), guide interferometer (Guide-1) and Guide-2 (a high-accuracy guide star-tracking telescope), and is a high-precision star-tracking telescope [6–9]. The SI has two sub-telescopes with 50 cm aperture and 6 m baseline length and is divided into two working modes: large FOV and small FOV. The field of view is 15° in the large field of view operation mode, and the astrometric accuracy is $4 \times 10^{-6 \prime\prime}$; the field of view is 2° in the small field of view operation mode, and the astrometric accuracy is $1 \times 10^{-6 \prime\prime}$ [6].

Although the modern OSA is flourishing, it must satisfy the optical path condition of geometric optics as well as the co-focal and co-phase of physical optics. The implementation of OSA is very difficult because the phase accuracy must be controlled within $\lambda/10$, and the PV values of the wave aberrations are within $\lambda/4$, which not only imposes extremely high requirements on the phase detection and correction controlling, but also requires very demanding stability of the mounting platform.

In order to break the limitation of the system aperture on spatial resolution enhancement in the passive non-coherent imaging regime and the shortcomings of the above-mentioned OSA imaging system, while taking into account the loose application requirements, we adopt an active coherent illumination imaging regime based on the Fourier ptychography (FP) imaging mechanism to extend the synthetic aperture radar (SAR) mode to far-field applications through FP phase recovery and spectral stitching techniques.

The Fourier ptychography microscopy (FPM) imaging technique was first proposed by Zheng, and a microscopic setup was successfully built in 2013 [10]. FPM is a powerful method to improve spatial resolution in microscopy [11–17]. In 2017, the concept of long-range synthetic aperture visible imaging (SAVI) based on the FP technique was formally proposed by Jason Holloway of Rice University, and for the first time, a long-range FP reflective imaging device for diffuse targets was built [18,19]. The device uses 532 nm laser active illumination with 1 m working distance and 2.5 m aperture diameter and captured multiple low-resolution (LR) images in the form of a $19 \times 19$ array to achieve a synthetic aperture of 15.1 mm [19].

Fourier ptychography imaging technology requires that the light filed has coherence and partial coherence, and it is more conducive to imaging everyday targets that scatter incident light in random directions [20–25]. However, none of the above imaging processes have been carried out to analyze the light filed coherence and the surface characteristics of rough targets. In order to fill the gaps in the above-mentioned research work, in this study, we analyze the coherence of the different laser modes and the properties of diffused objects with diverse materials.

In this study, firstly, we analyze the imaging theory of the optical system, which can help us understand the reflective macroscopic FP imaging process more effectively. Secondly, to achieve all-day observation or imaging requirements, we use a near-infrared band for imaging, and we analyze the coherence of the laser with different modes and the characteristics of diffuse reflectivity for five materials, which is the vital prerequisite to realize remote sensing imaging and ensure the effectiveness of the FP phase reconstruction algorithm. Thirdly, based on the above analysis, we explore the active coherent synthetic aperture high-resolution imaging technique that can break the optical diffraction limit in the far field and further extend the application of reflective FP imaging technology using camera scanning for diverse materials of rough surfaces with a working distance of 3310 mm as an example, which is the longest working distance of the current reflective

FP imaging system. Meanwhile, based on the scattering imaging of diffused targets, the resolution enhancement ratio is 1.97× of the real target by the average speckle size calculated using the normalized autocorrelation function and 2× resolution improvement ratio of the USAF resolution target, which will be the new method for the imaging quality evaluation. Finally, we summarize the results and discuss the limitations and the influential factors in long-range FP imaging research.

## 2. Imaging Theory

In order to make it easier to understand how the light field propagates, based on the far-field reflective imaging model, built a simplified sketch, as shown in Figure 1. Consider the light field distribution of the diverging spherical wave light field with wavelength $\lambda$ emitted from the point source after propagation distance $z_1$ as [26]

$$\vec{U_1}(x_1, y_1) = \frac{a_0}{z_1} \exp(jkz_1) \exp\{j\frac{k}{2z_1}[(x_1 - x_0)^2 + (y_1 - y_0)^2]\} \qquad (1)$$

where $\vec{U_1}(x_1, y_1)$ indicates the light field distribution on the front surface of the lens $f_1$. $a_0$ indicates the amplitude per unit distance condition. $k = \frac{2\pi}{\lambda}$ is the wave number. If we set $(x_0, y_0)$ as the origin of the coordinate, and $(x_1, y_1)$ is the space coordinate on the plane of the front surface of the lens $f_1$. Equation (1) is given as

$$\vec{U_1}(x_1, y_1) = \frac{a_0}{z_1} \exp(jkz_1) \exp[j\frac{k}{2z_1}(x_1{}^2 + y_1{}^2)] \qquad (2)$$

After the modulation effect of the lens $f_1$, the light field distribution on the rear surface of the lens $f_1$ gives,

$$\vec{U_2}(x_2, y_2) = \frac{a_0}{z_1} \exp(-jkz_1) \exp\{-j\frac{k}{2}\frac{1}{z_1 + f_1}[(x_2 - x_1)^2 + (y_2 - y_1)^2]\} \qquad (3)$$

where $(x_2, y_2)$ is the spatial coordinate on the plane of the rear surface of the lens $f_1$. The light field expressed by Equation (3) reaches the front surface of the object after a free propagation distance $z_{21}$ in space, assuming that the surface reflectivity of the object is $\vec{R}$. At this time, the light field of the object after reflection is expressed as

$$\vec{U_3}(x_3, y_3) = \frac{\vec{R}}{\sqrt{j\lambda z_{21}}} \int \int_{-\infty}^{+\infty} \vec{U_2}(x_2, y_2) \exp\{j\frac{k}{2z_{21}}[(x_3 - x_2)^2 + (y_3 - y_2)^2]\} dx_2 dy_2 \qquad (4)$$

For specular reflective targets, $\vec{R} = 1$ is the same at every location in space, however, for diffused objects, the reflectivity $\vec{R} = \vec{R}(x, y)$ varies with the location in space. Similarly, the reflected wave $\vec{U_3}(x_3, y_3)$ propagates over a sufficiently large distance $z_{21}$ toward the aperture plane, which satisfies the far-field Fraunhofer approximation. Assuming that the pupil function denotes $\vec{P}(u, v)$, after the modulation of the optical pupil function, the light field $\vec{U_f}(u, v)$ is related to the field $\vec{U_3}(x_3, y_3)$ through Fourier transform and the light field distribution on the rear surface of the lens $f_2$ is

$$\vec{U_f}(u, v) = \vec{P}(u, v) \bullet F\{\vec{U_3}(x_3, y_3)\}. \qquad (5)$$

where $(u, v)$ is the frequency coordinate. Since the current detector can only record intensity information lost phase information, the light intensity on the image plane of the detector $I_{img}(x_{img} y_{img})$ is expressed as

$$I_{img}(u_{img}, v_{img}) = |F\{\vec{P}(u, v) \bullet F\{\vec{U_3}(x_3, y_3)\}\}|^2. \qquad (6)$$

where $(x_{img}, y_{img})$ is the spatial coordinate of the image plane of the detector. In remote sensing imaging, the laser point source generates the coherent light, and the camera must be located on the same side; it belongs to the reflection imaging model. The study gives an equivalent imaging model of aperture scanning which is suitable for remote sensing applications. Firstly, the position relationship of the light source, camera and target in Figure 1 definitely satisfies the object-image conjugate relationship required by geometric optics; secondly, it is well known from Fourier optics theory that the remote sensing distance fully satisfies the far-field Fraunhofer diffraction approximation [19,26], and the objective complex amplitude after laser active illumination will be diffracted and propagated by the light field, and the diffracted light field arriving at the aperture is exactly the Fourier spectrum of the target; thirdly, the diffraction light field coincides with the focal plane of the detector, and only then the low-resolution intensity images recorded by the detector meets the requirement of FP super-resolution reconstruction; finally, adopting an external aperture structure, the whole camera movement is naturally equivalent to the implementation of aperture scanning for the spectral plane of the target. Therefore, we believe that the camera scanning Fourier ptychographic imaging model has the potential to realize remote sensing applications.

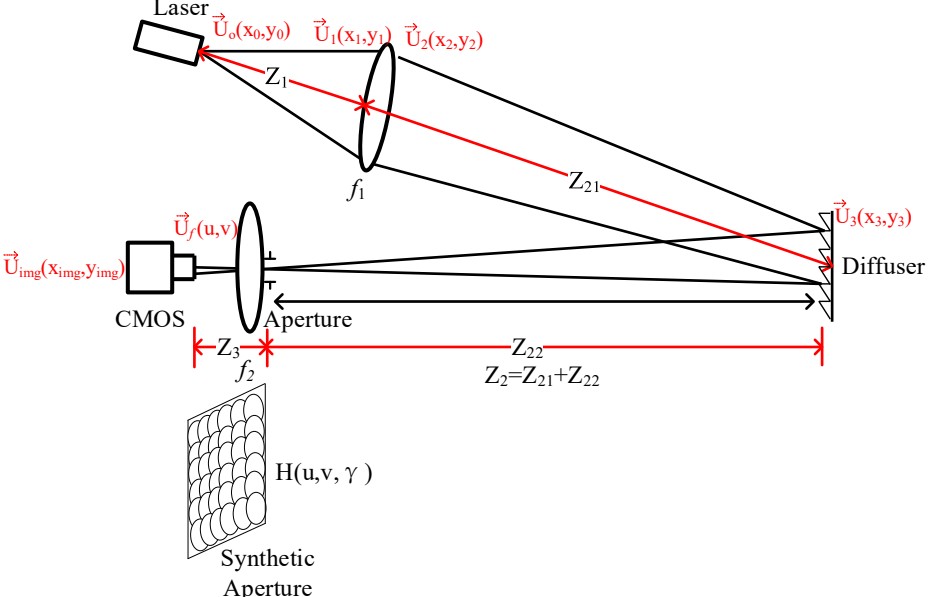

**Figure 1.** The geometric model of the reflective Fourier ptychographic optical imaging system.

Much work has been reported on the simulation of Fourier ptychography far-field imaging systems [18,19], the details of which will not be discussed in this study.

## 3. Experiment

In order to ensure the success of the experiment and the effective recovery of the high-resolution images from the acquired data, we need to put forward the corresponding requirements for the coherence of the laser and the target surface properties in the experimental process [27,28]. In this study, we briefly analyze the above two aspects and clarify the a priori conditions for successful experimental realization.

### 3.1. Requirements of Laser Light Source

As the laser active lighting system emits light and the return signal is propagated in the air, the background radiation, transmittance, visibility and other characteristics of different distances of the lighting area are not the same, which is necessary to adjust the laser emission power to ensure that the illumination is relatively uniform and consistent [28].

### 3.1.1. Single-Mode Laser Light

Generally speaking, the higher the pump power, the more oscillation lines of different frequencies the output will contain. Thus, we can use these to distinguish between lasers operating with single-mode oscillation and multimode type of oscillation [28].

Single-mode lasers are fully coherent light. The ideal single-mode oscillation of laser light is a monochromatic wave with known amplitude $S$, known frequency $\bar{v}$ and fixed but unknown absolute phase $\phi$; then, the optical field distribution of the single-mode oscillating laser is shown as [28]

$$u(t) = S\cos(2\pi\bar{v}t - \phi) \tag{7}$$

In practice, we do not know the absolute value of the phase $\phi$, which can be treated as random variables and is uniformly distributed in the interval as $(-\pi, \pi)$.

In Figure 2, the 2D light field intensity distribution image of the single-mode fiber laser shows that the single-mode fiber laser exiting the light field is closer to the Gaussian model; meanwhile, the actual single-mode laser emits a light beam pattern as shown in Figure 3. The two figures shows that the single-mode laser emits a uniform and consistent light beam.

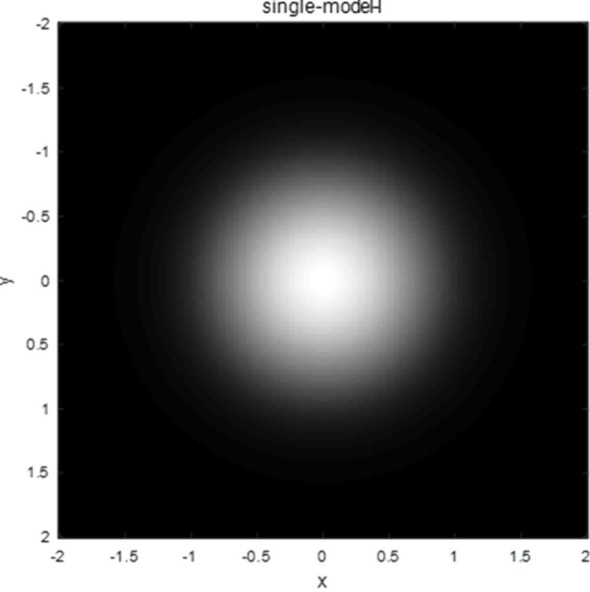

**Figure 2.** The 2D light field intensity distribution image of the single-mode fiber laser.

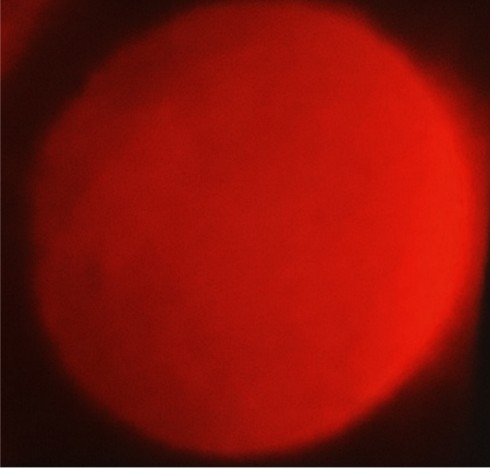

**Figure 3.** The real image of light beam pattern emitted by an actual single-mode fiber laser.

### 3.1.2. Multimode Laser Light

The modes are generally divided into longitudinal and transverse mode structures or both. For a multimode oscillating laser, the output optical field distribution at the point $P$ is shown as [27–29]

$$u(P,t) = \sum_{n=1}^{N} S_n(P) \cos(2\pi \nu_n t - \phi_n(P,t)) \tag{8}$$

where $N$ is the number of the modes; $S_n(P)$ and $\nu_n$ are the amplitude and center frequency of the $n$th mode respectively, and $\phi_n(P,t)$ is the phase variation associated with the $n$th mode, including the constant starting phase $\phi_n$. From the above Equations (7) and (8), we can find that the output light distribution of single-mode lasers is synthesized from fixed amplitude, frequency and phase, which has a more coherent nature. However, multimode laser light is composed of multiple modes with different frequencies and phases, and the output light field distribution is the result of the summation of the light field at each frequency and phase, which is less coherent compared to single-mode lasers. In addition, because there is no coherence in non-degeneracy modes in multimode lasers, for example, there are multiple linear polarization (LP) modes in multimode fibers, and each LP mode includes a different degeneracy, for example, $LP_{01}$ mode, containing two $HE_{11}$ modes with coherence has a degeneracy of 2, while $LP_{11}$ mode has a degeneracy of 4, containing one $TE_{01}$ mode, one $TM_{01}$ mode and two $HE_{21}$ modes, which are coherent to each other. However, there is no coherence between $LP_{01}$ and $LP_{11}$ modes. Therefore, the light emitted from the multimode fiber is partially coherent and the more the modes, the less the coherence. Meanwhile, the farther the beam travels in the air, the worse the coherence is. The beam quality is also inversely proportional to the number of modes, and the relationship between the beam quality and the number of modes is given [28,29].

$$M_x^2 = \sum_{m=0}^{\infty} \sum_{n=0}^{\infty} (2m+1)|C_{mn}|^2 \tag{9}$$

$$M_y^2 = \sum_{m=0}^{\infty} \sum_{n=0}^{\infty} (2n+1)|C_{mn}|^2 \tag{10}$$

where $M_x^2$ and $M_y^2$ denote the beam quality in the $x$ direction and $y$ direction, respectively. $C_{mn}$ is the weight factor. When the $M^2$ factor is closer to 1, the beam quality is better and the corresponding divergence angle is smaller. $m$ and $n$ denote the number of modes in the $x$ direction and $y$ direction, respectively, and from Equations (9) and (10), it can be seen that the greater the number of modes, the worse the beam quality [28]. The intensity distribution of the optical field of a multimode fiber with a different number of modes is shown in Figure 4. While the multimode fiber laser has two symmetrical main flaps for the $LP_{11}$ mode, the sum of the $LP_{11}$ and $LP_{12}$ modes has symmetrical side flaps in addition to the two symmetrical main flaps. The actual multimode fiber laser spot is shown in the Figure 5. The light beam quality is poor, as seen in the pattern, while the effective recovery of the experiment cannot be guaranteed by using this laser for diffused imaging experiments.

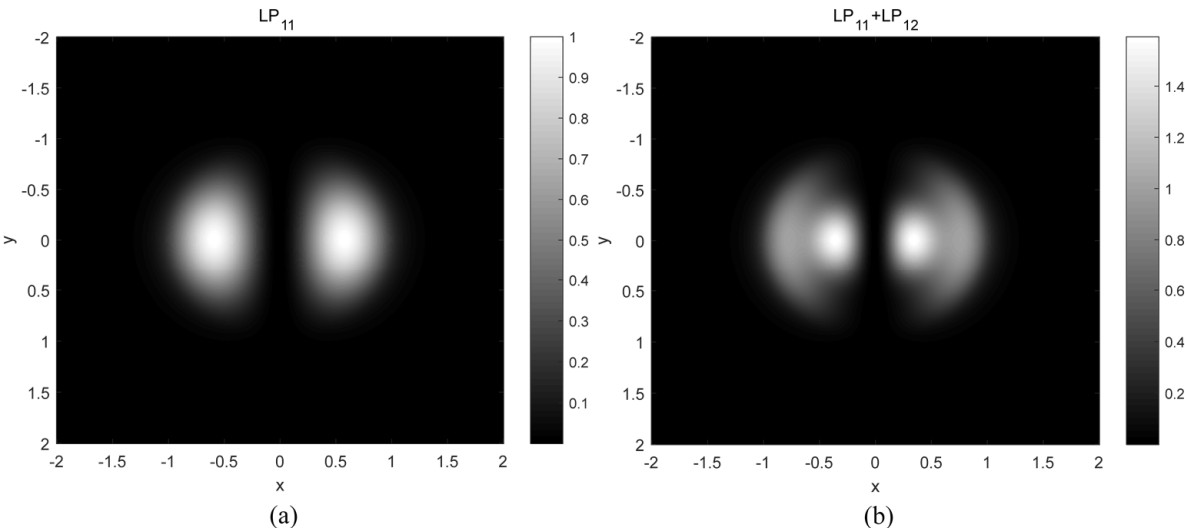

**Figure 4.** (**a**) The simulation image of $LP_{11}$ mode light field intensity distribution; (**b**) the simulation image of the sum of $LP_{11}$ and $LP_{12}$ mode light field intensity distribution.

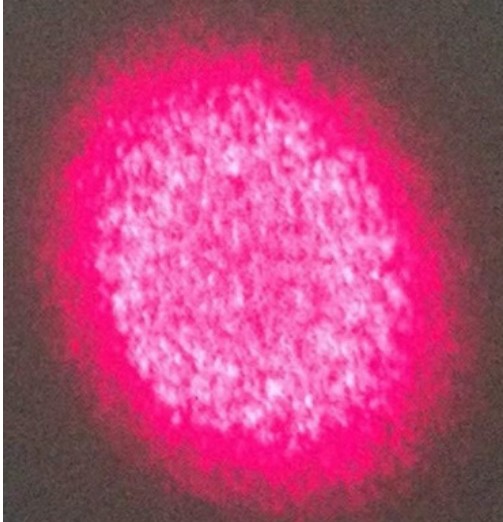

**Figure 5.** The real image of light beam pattern emitted by an actual multi-mode fiber laser.

When a fiber laser illuminates the practical rough surface, the roughness of the target can destroy the coherence of the laser or even make it disappear, leading to the failure of the imaging algorithm of the FP; however, the degree of destruction is not the same for single-mode and multimode laser light. Therefore, through the above analytical coherence of the single-mode fiber laser and multimode fiber laser, the choice of a single-mode fiber laser with great coherence can ensure the effective recovery of the FP recovery algorithm. The laser active lighting system is shown in Figure 6.

Our experimental system consists of the laser emission device, the object and the imaging device (imaging lens and CMOS camera). The laser emission device contains three parts: the laser, focusing lens1 and focusing lens2. The object is illuminated by the laser with a 976 nm single-mode fiber near-infrared semiconductor. The coherent light beam directly illuminates focusing lens1 ($f_1$= 160 mm), lens2 ($f_2$= 2000 mm) and the object. Lens1 is used to compress the laser divergence angle, and lens2 is used to image the laser point light source. Similar to the previously reported long-distance FP setup [19], our imaging system consists of a high performance camera (MVC10KMF-M00), coupled with a 180 mm Nikon photographic lens (Tokyo, Japan, AF NIKKOR 180MM F/2.8, IF-ED) mounted on a motorized 2D stage (GCD-20330M, see Figure 6), which was

operated manually. The guide rail of the translation table is 300 mm with 1 μm accuracy. We successively obtained low-resolution images of 81 different positions with 9 × 9. The vertical and the horizontal positions are the same by 10 mm, so neighboring images overlapped and the overlap ratio is 83%.

Imaging lenses with $f_{img} = 180$ mm can be easily (and cheaply) manufactured to have a proportionally large aperture. In order to simulate building a cheap lens with a long focal length and small aperture, we create 60 mm diameter aperture. The low-resolution images are acquired by moving the camera with equidistant positions between the two adjacent images to synthesize the large aperture via spectrum stitching.

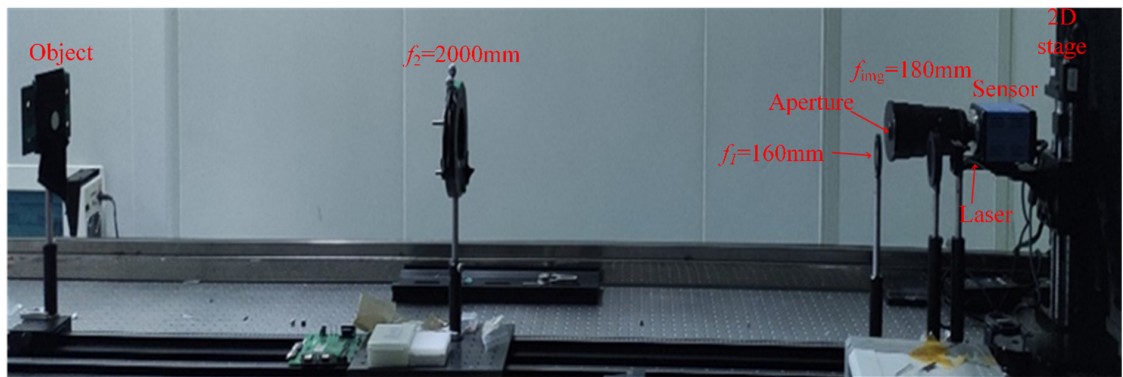

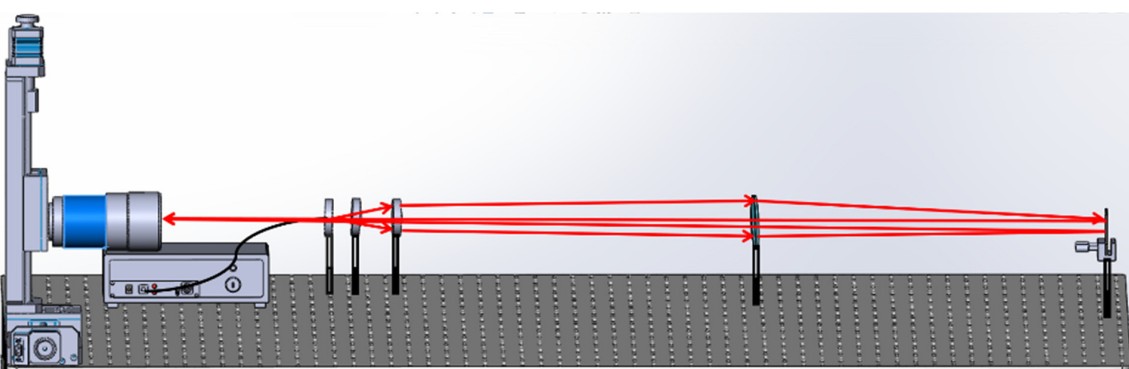

**Figure 6.** The sketch of reflective Fourier ptychography imaging setup. The single-mode fiber infrared semiconductor laser with wavelength of 976 nm emanates coherent light to illuminate the target. The aperture acts as a low-pass filter of Fourier transform, and the information undergoes inverse Fourier transform. The camera sensor receives the intensity measurements at different positions.

### 3.2. Qualitative Analysis of the Effect of Target Surface Properties on Laser Coherence

From the theory of light field propagation, it is known that the reflected light from the surface of an object will have different reflective components due to the influence of surface roughness, reflectivity and other factors. In 1991, Wolff and Boult, after an in-depth analysis of the factors influencing the surface of an object, combined with Fresnel's law to provide a detailed classification of the reflected light components of non-Lambertian surfaces into four categories [29]. In this study, we only focus on specular and diffused reflections.

The specularly reflected light is formed when the wavelength size of the incident light is much smaller than the size of the surface of the object, and the light is highly directional and can usually only be observed in a specific direction [29], as shown in Figure 7. Diffusely reflected light reflects the scattering and absorption of the incident light in random directions. For the viewer, it can be used to distinguish the color information of the object and carry the rich surface shape information of the object. This means that the diffusely and specularly reflected light are independent of each other, and according to this property, the low-resolution intensity image is expressed in the following form [29].

$$I_{image} = I_{diffuse} + I_{specular} \tag{11}$$

where $I_{image}$ is the total light intensity image on the image plane; $I_{diffuse}$ and $I_{specular}$ are the diffuse and specular reflection intensity images of the target surface, respectively.

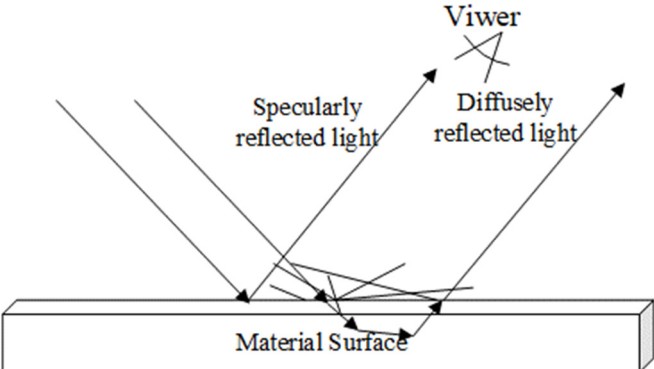

**Figure 7.** The sketch of specular and diffuse reflections on the surface of an object.

Fourier ptychography imaging is a coherent or partially coherent imaging technique, and five different materials were chosen for the experiments in order to verify the property, including a one-yuan RMB bill, a resolution target with pottery and porcelain, a one-yuan coin with a nickel-plated steel core, a fifty-cent coin with a steel core copper-plated alloy and a ten-cent coin with aluminum alloy for acquisition and HR reconstruction, respectively. As shown in Figures 8 and 9, the one-yuan RMB bill and ceramic resolution target could not achieve recovery under this experimental condition, but good reconstructed results were achieved for metallic materials such as the one-yuan coin with steel core nickel-plated material, fifty-cent coin with steel core copper-plated alloy material, and ten-cent coin with aluminum alloy material. In addition to these three metallic materials, we also tested keys, circuit boards, and other metallic materials, all of which can recover satisfactory super-resolution results.

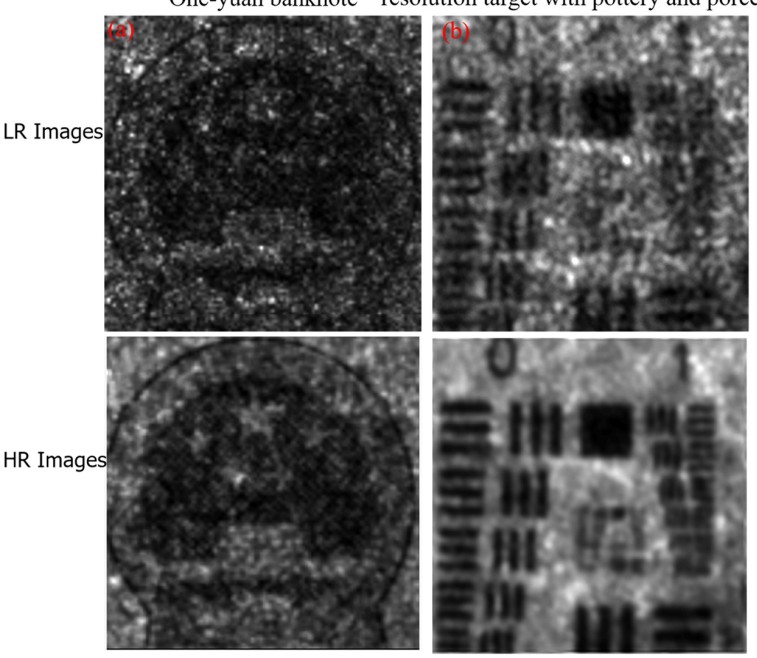

**Figure 8.** The reconstructed images with low resolution of non-metallic materials. (**a**) is a one-yuan RMB bill; (**b**) is a resolution target with pottery and porcelain.

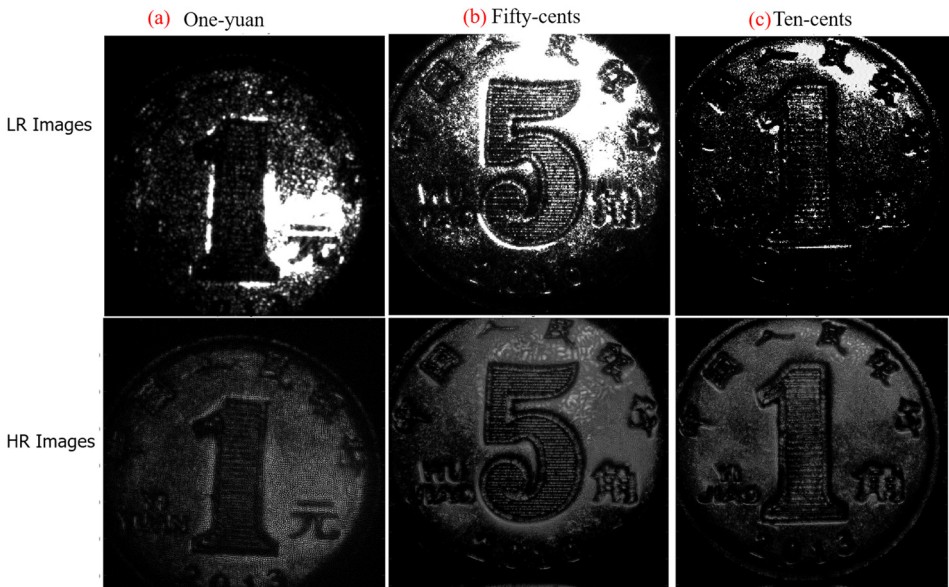

**Figure 9.** The reconstructed images with high resolution of metallic materials. (**a**) is a one-yuan coin with a nickel-plated steel core; (**b**) is a fifty-cent coin with a steel core copper-plated alloy; (**c**) is a ten-cent coin with aluminum alloy.

Our explanation for this phenomenon focuses on two aspects: Firstly, for banknotes and ceramic targets, the coherence of the laser is completely destroyed by the rough surface of the target after long-distance propagation, and there is no coherence relationship between the low-resolution images, which leads to the high-resolution images after iteration being only the result of averaging all the low-resolution images and no obvious resolution improvement; while for all the coins, these metallic materials belong to mixed specular and diffused targets, and the laser irradiation to the target surface still retains some coherence or is partially coherent, so that there is still a certain coherence relationship between the low-resolution images, which can obtain the obvious resolution improvement after the recovery via the iterative process. Secondly, these surfaces of banknotes and ceramics do not have a strong reflection effect on 976 nm band light, and neither do the target surfaces of these materials have a certain absorption effect on this band, which means the target information and background information cannot be completely separated and causes invalidity of the reconstructive algorithm.

Above this, the experiment has two requirements for the laser and the target surface. Firstly, when the laser irradiates the diffused target surface, the roughness of the target surface will destroy the coherence of the laser to some extent. For multimode fiber lasers, due to the light field being partially coherent, the rough surfaces are more likely to destroy the coherence of the light field compared with single-mode lasers, and the FP phase algorithm requires a certain coherence of the light field to recover the high-resolution image, thus, the single-mode laser can be used to ensure the coherence of the light field more effectively. At the same time, the beam quality of multimode fiber lasers is poorer than that of single-mode lasers, which cannot uniformly and consistently illuminate the target surface.

Secondly, in this experiment, in order to be able to verify the all-day imaging requirements, an infrared 976 nm light was used to illuminate the target surface; a high diffuse reflectivity is required on the target surface alone to achieve reconstruction.

According to Ref. [19], whether it is a single-mode fiber laser or a multimode fiber laser, as long as the coherence of the light field can be maintained throughout the imaging process and the high reflectivity of the experimental target can be guaranteed, it not only assures the successful implementation of the far-filed FP experiment, but also determines the effectiveness of the recovery algorithm. However, in the long-range experiment, the working distance, the surface characteristics of imaging target and the influence of various

factors on the coherence should be considered. Therefore, the selection of the light source has become a particularly important segment. In this study, in order to achieve effective long-distance imaging and ensure the follow-up research work, we chose a single-mode fiber laser as the light source with great beam quality and strong coherence.

### 3.3. Experimental Analysis

To elaborate on the above results, we imaged the fifty-cent coin with an experimental system at a working distance of 3310 mm as an example. Via analyzed the scattering size, the average speckle size before and after the recovery algorithm was obtained to quantify the diffuse inverse target enhancement ratio, and the comparison of the resolution enhancement ratio before and after the recovery of the specular resolution target was used to verify the validity of the above scattering analysis.

#### 3.3.1. Diffused Object Imaging

The concrete parameters of the imaging system are shown in Table 1. The target was placed 3310mm away from the pupil plane of imaging lens. Throughout the experiment, a total of 81 low-resolution images were collected and the scanning step is 10 mm with a 83% overlap ratio between two adjacent images.

**Table 1.** Parameters of the imaging system.

| Parameters | Values |
| --- | --- |
| Wavelength | 976 nm |
| Aperture diameter | 60 mm |
| Pixel size | $1.67 \times 1.67$ μm |
| Number of pixels | $3664 \times 2748$ |
| Working distance | 3310 mm |

In Figure 10, the diffused object is illuminated with coherent light: an image of a fifty-cent coin is presented. A single captured image for the object is unrecognizable because of the degradation from the speckle patterns and diffraction blur. After reconstruction using FP, images are recognizable and fine details can be clearly observed, as shown in Figure 11. The number of adjacent pixel shifts corresponding to a sampling interval of 10 mm is 256 pixels. From the perspective of the recovery effect, the diffuse target is reconstructed via FP super-resolution. During phase retrieval [12,30,31], a high-resolution estimate of the phase emanating from the real scene is recovered. As expected for a diffused object, the phase exhibits a random profile in the range $[-\pi, \pi]$.

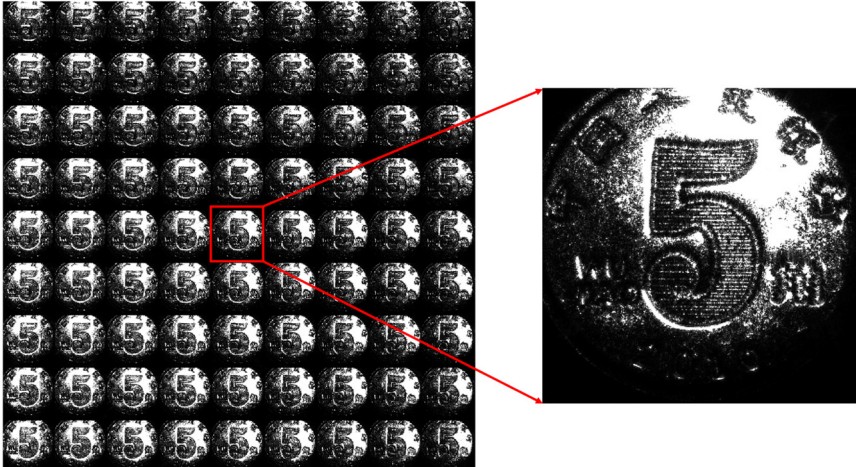

**Figure 10.** The low-resolution image array with 81 images and the central image of a fifty-cent coin.

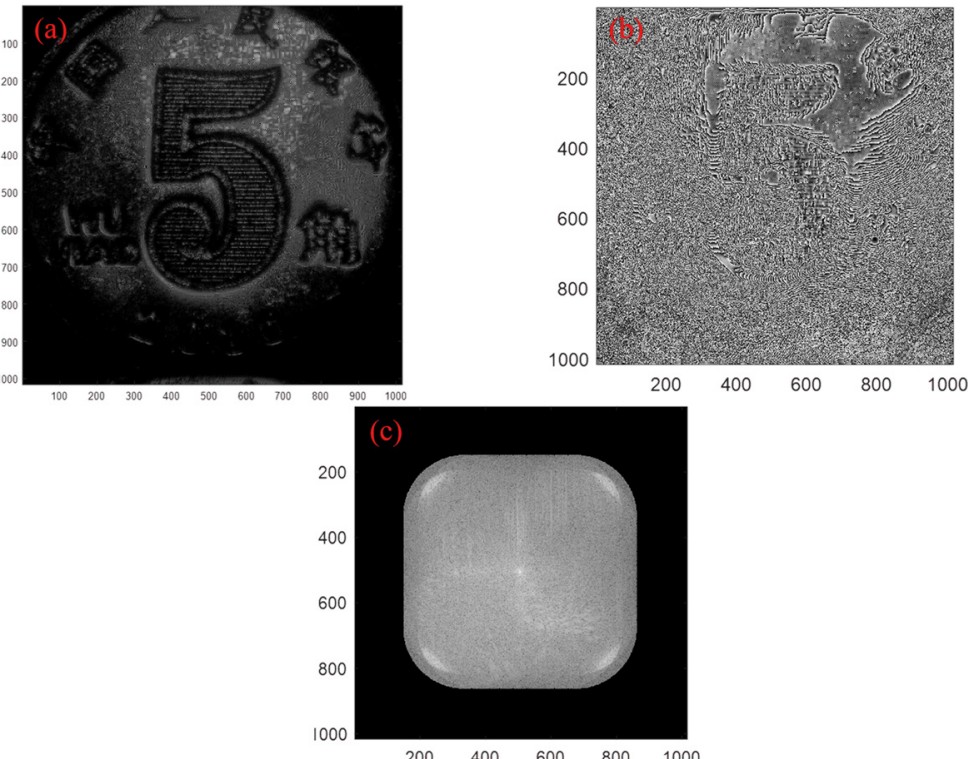

**Figure 11.** The reconstructed high-resolution results of a fifty-cent coin in this experiment. (**a**) is the reconstructed high-resolution intensity image of a fifty-cent coin; (**b**) is the reconstructed phase image of a fifty-cent coin; (**c**) is the synthetic Fourier spectrum image of the reconstructed high-resolution image.

### 3.3.2. Speckle Imaging Analysis

In order to further analyze the diffused target, using the method mentioned in Refs. [20,21], we calculated the speckle size for the original low-resolution image and the reconstructed high-resolution image by selecting three identical positions, as shown in Figure 12. The correlation width is measured by the full width of the curves at their half-maxima (FWHM) [27]. The coherence width in the horizontal direction is denoted as HFWHM, and similarly, the coherence width in the vertical direction is denoted as VFWHM [24,25]. The size of the speckle area (speckle size) is expressed as the product of the coherence widths in both directions. Thus, in this study, we defined the ratio of speckle size between the high-resolution image and raw image as the resolution improvement ratio. We randomly and respectively selected 10 positions in the above raw low-resolution image and the high-resolution images to obtain the resolution enhancement under different positions, as shown in Figure 13. Meanwhile, to improve the accuracy of the calculation results, the average gains of the resolution were $1.97\times$ by averaging multiple calculated values, and the reconstructed image has higher resolution, more consistent dynamic range, and has essentially the same resolution improvement gains as the USAF resolution target.

The results presented in this study experimentally confirm the feasibility of reflective macroscopic FP synthetic aperture, which can enhance resolution and reduce speckle size. We can not only judge the resolution improvement ratio intuitively using the USAF resolution target but also using the calculation of the average speckle size, which is more universal to the practical object without the dependence on the USAF resolution target.

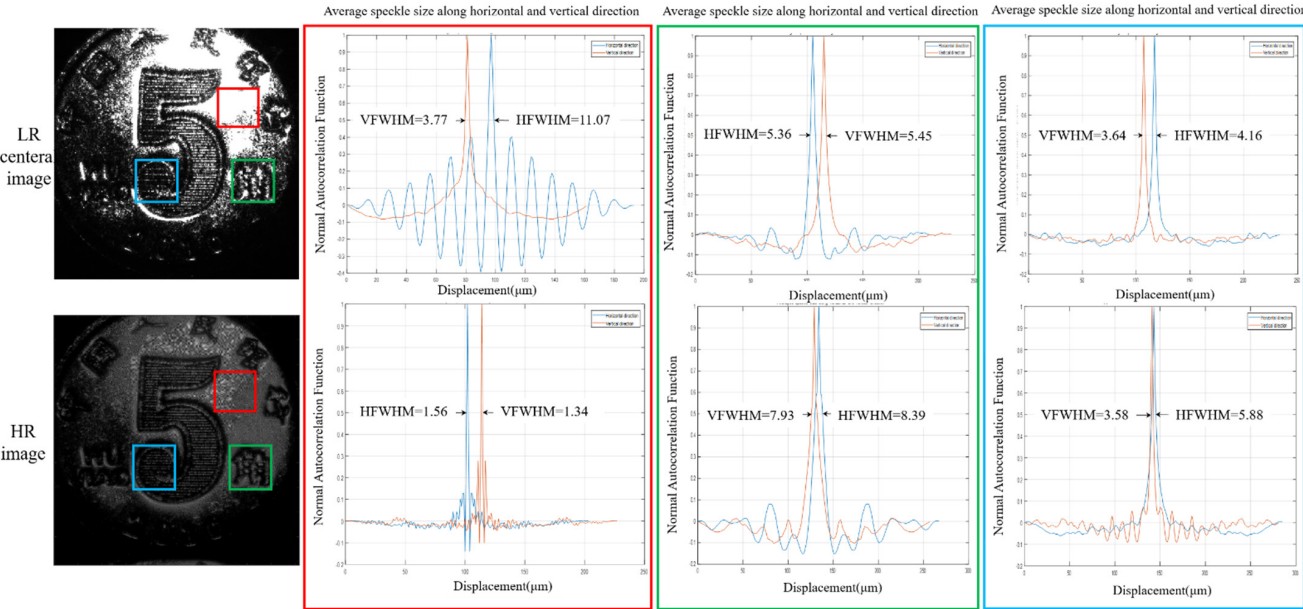

**Figure 12.** The results of average speckle sizes of the LR central image and the HR image at three random positions.

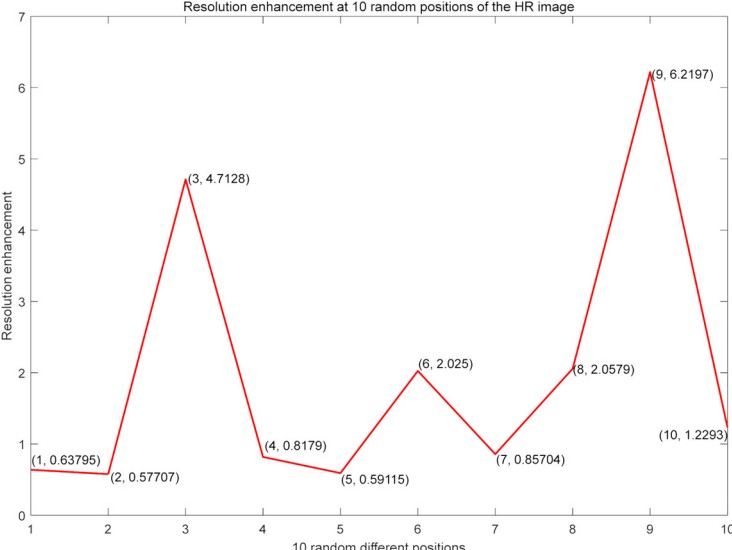

**Figure 13.** The plot of the resolution enhancement ratio at ten random different positions of the LR central image and the HR image above.

### 3.3.3. Verification Results with USAF Resolution Target

In order to quantify the resolution improvement performance of the reflective macroscopic Fourier ptychographic synthetic aperture imaging system, we imaged positive USAF chrome-on-glass with anti-reflection coating. The target is imaged to retain the high-resolution feature characteristic of the resolution chart. An example of the captured images is shown in Figure 14. Figure 14a shows an original sequence of images composed of 81 low-resolution images. Figure 14b shows a central image, the resolution of which is limited to 8 lp/mm or a bar width of 62.5 μm (Group 3 Element 1).

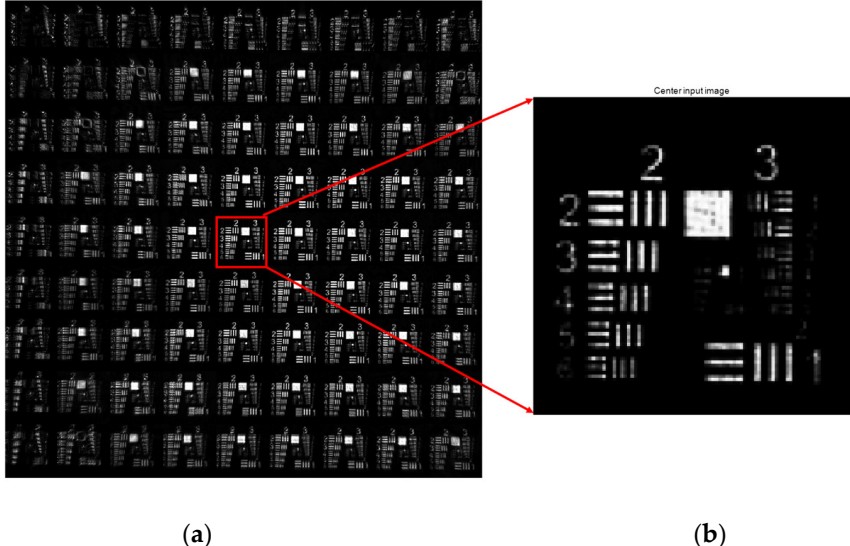

(**a**)                                                        (**b**)

**Figure 14.** The low-resolution images acquired by the laboratory macroscopic far-field Fourier ptychography setup with scanning pupil of 10 mm. (**a**) is the low-resolution image array with 81 images; (**b**) is the 41st center input image with low-resolution.

In Figure 15, high-resolution images are reconstructed by only using an 81 image data set with a sampling interval of 10 mm with moving synthetic aperture. The reconstructed intensity image of the USAF resolution target is shown in Figure 15a. Figure 15b is the recovered phase image, and Figure 15c shows the synthetic aperture spectrum of the high-resolution image. The resolution increased to 16 lp/mm or a bar width of 31.25 μm (Group 4 Element 1). The resolution improvement is 2×.

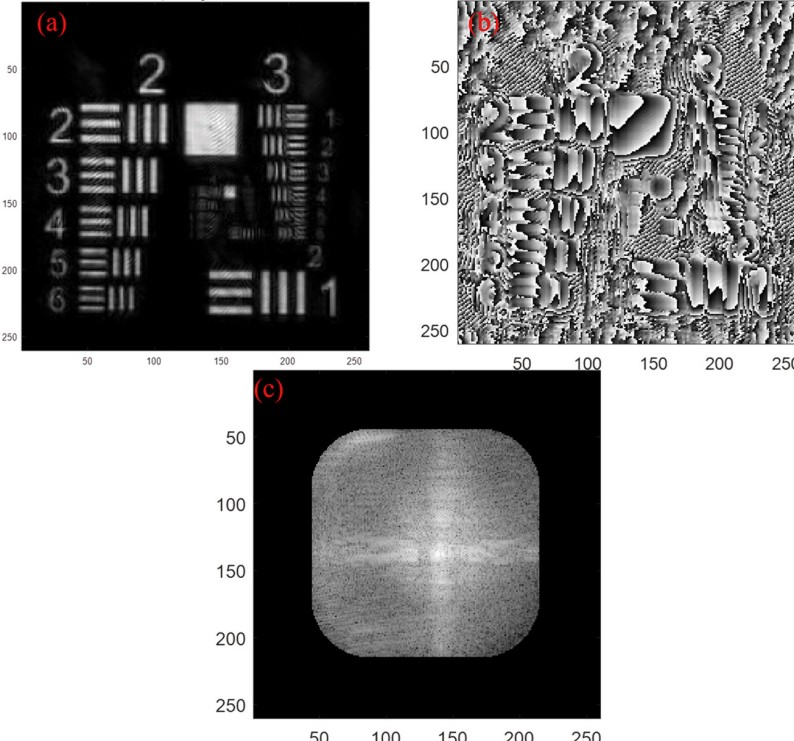

**Figure 15.** The reconstructed high-resolution results of USAF target in this experiment. (**a**) is the reconstructed high-resolution intensity image of USAF target; (**b**) is the reconstructed phase image of USAF target. (**c**) is the synthetic Fourier spectrum image of the reconstructed high-resolution USAF image.

The resolution improvement ratio of the USAF resolution target highly coincides with the resolution improvement ratio of the diffused object calculated using the speckle size, which demonstrates the validity of the calculation method to estimate the resolution improvement multiplier of the super-resolution reconstruction of rough targets.

## 4. Discussion

Starting from the speckle formation mechanism of active coherence imaging, a quantitative evaluation method is proposed in this study, which provides a means of high application value for the quantitative evaluation of the resolution improvement ratio of Fourier ptychography remote sensing imaging for real scenes. Speckles have a random nature and no spatial reference; therefore, the average of multiple calculations can be used to estimate the speckle size more effectively.

Camera scanning synthetic aperture imaging technology under the framework of FP is studied. Through the comparative analysis with passive incoherent optical synthetic aperture imaging technology and other active imaging technologies, the advantages of the FP synthetic aperture imaging technique in far-field high-resolution imaging in this study are highlighted.

### 4.1. FP and Passive Incoherent Optical Synthetic Aperture Imaging technology

Improving the resolution of space optical cameras is an unremitting goal in the field of high-resolution earth observation. In the traditional passive incoherent imaging system, the improvement of resolution depends on the increase of aperture, but the high manufacturing and launch costs limit the development and application of large aperture systems. Although synthetic aperture imaging technology can bypass the limitation of large-diameter mirrors, the imaging systems must meet strict confocal and co-phase conditions between multiple sub-apertures, which poses extremely high requirements for phase detection, correction control and platform stability. The technology applied in this study breaks the limitation of a passive incoherent imaging system, realizes the effect of synthetic aperture by means of a single, small aperture scan and overcomes the confocal and co-phase difficulties between multiple sub-apertures, and the traditional passive imaging methods miss the phase information. As an advanced phase retrieval technology, FP integrates spectrum shift, phase retrieval, spectrum stitching and aperture synthesis by means of coherent light illumination. By using the low-resolution intensity image with redundant spectrum information for fusion reconstruction, the high-resolution intensity image and phase information can be synchronously reconstructed. It breaks through the diffraction limit of an optical system and achieved success in this study in far-field high-resolution, high-throughput imaging.

### 4.2. FP and Other Active Coherent Imaging Technologies

In order to break the limitation of a passive, incoherent imaging system, a variety of new, active, coherent imaging technologies emerge endlessly, which contain synthetic aperture digital holography (SADH), intensity correlation imaging technology (ICIT), speckle correlation imaging technology (SCIT) and FP. The main differences between these imaging technologies lies in whether the phase information can be recovered and the difficulty of phase recovery. A comparison of technical characteristics among these four imaging technologies is shown in Table 2. The analysis results show that FP has the advantages of high resolution, high signal-to-noise ratio (SNR), no interference and is a simple experimental device. At the same time, it can reconstruct the phase information as accurately as digital holography and correct aberrations, suppressing the atmospheric disturbance effectively. With increasing the laser illumination distance, improving the optical aperture and using high-sensitivity detectors, it is entirely possible for FP to further expand the working distance to achieve high-resolution imaging from the current meter-level to spatial remote sensing.

**Table 2.** Comparison of technical characteristics among several, active, coherent imaging technologies.

|  | Realization Conditions | Resolution in the Same FOV | Phase Retrieve | Time Resolution | SNR | Working Distance | Effect of Atmospheric Disturbance |
|---|---|---|---|---|---|---|---|
| SADH | Strict (Need interference) | HR (<5×) | Accurate | High | High | Long-distance | Strong |
| ICIT | Loose | LR | No | Low | Low | Long-distance | weak |
| SCIT | Loose (No interference) | LR | No | Low | Low | Long-distance | weak |
| FP | Loose (No interference) | HR (>5×) | Accurate | High | High | Long-distance | weak |

## 5. Conclusions

In this paper, firstly, a mathematical model of reflective FP imaging for the far field is proposed and studied based on the Fraunhofer far-field diffraction mechanism, which establishes a theoretical foundation for the application of Fourier ptychography synthetic aperture imaging technology in practical spatial remote sensing. Secondly, we systematically analyze the influence of laser coherence in different modes and the surface properties of diverse materials for diffused targets. The analysis results prove that using a single-mode fiber laser as the illumination source and metal materials with high diffused reflectivity as the experimental targets can ensure the validity of the experimental results. Finally, we emulate camera scanning with a single camera attached to a X-Y translation stage, and an experimental system with a working distance of 3310 mm is used to image a fifty-cent coin and USAF target. A quantitative evaluation method is proposed for the resolution enhancement ratio of the rough targets, which is based on the statistical properties of the speckle. The normalized autocorrelation function is used to calculate the average speckle size, and the ratio of the average speckle sizes before and after super-resolution can quantitatively characterize the resolution improvement ratio. The experimental results prove the great potential of Fourier ptychography synthetic aperture imaging technique in far-field high-resolution imaging.

**Author Contributions:** Conceptualization, M.Y. and H.Z.; methodology, M.Y.; software, M.Y.; validation, M.Y. and H.Z.; formal analysis, M.Y. and Y.W.; investigation, M.Y., X.F., Y.W. and H.Z.; resources, M.Y.; data curation, M.Y. and Y.W.; writing—original draft preparation, M.Y.; writing—review and editing, M.Y. and H.Z.; visualization, M.Y.; project administration, X.F. and H.Z.; funding acquisition, X.F. and H.Z. All authors have read and agreed to the published version of the manuscript.

**Funding:** This research was funded by the Major Project on the High-Resolution Earth Observation System (GFZX04014307).

**Acknowledgments:** The authors thank Baoli Yao, An Pan and Jing Wang for their help with the experiments and discussion.

**Conflicts of Interest:** The authors declare no conflict of interest.

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
