# Peer review of "Experimental Study on the Exploration of Camera Scanning Reflective Fourier Ptychography Technology for Far-Field Imaging"

_remotesensing, doi:10.3390/rs14092264_

Round 1

Reviewer 1 Report

Fourier ptychography imaging is a powerful phase retrieval method that can be used to realize super-resolution. In this study, the authors established a mathematical model of long-distance camera scanning base on reflective Fourier ptychography imaging to analyze the coherence of the different laser modes and the properties of diffused objects with diverse materials. This study is important of the further development of the computer vision, surveillance, remote sensing, et al.

Main contents:

1.The structure of the paper is bad. In Discussion, the authors do not give a good evaluation of their work, but provide abundant future ideas.
2.The conclusions of the work should be rearranged.
3.The authors provide a lot of Figures, while the description of the Figures is rather poor. Also, the descriptions are subjective. In many places, the authors say “we can see …”. This makes the paper is hard to understand.
4.The description of the main parameters in the Equations are absent.
5.There exist many expression problems. E.g.

(1)Line 24: the dot is absent.

(2)Line 60: “is 4×10-6″”

(3)Line 96: “based above analysis,”

(4)Line 225: “The function of lens1 is to compress the laser divergence angle, and lens2 is…”.

Please be careful to revise the expression problems.

Reviewer 2 Report

Dear authors,

I found your article is very interesting and scientifically sound .  The article is well written and structured. One comment I have is your discussion part is a bit shallow and I suggest somehow to rework on it.

Best regards

Reviewer 3 Report

The reviewer is really grateful to the authors for such an outstanding research and article about experimental study on the exploration of camera scanning reflective Fourier ptychography technology for far-field imaging. Fourier ptychography imaging is a powerful phase retrieval method that can be used to realize super-resolution. In this study, the authors establish the mathematical model of long-distance camera scanning base on reflective Fourier ptychography imaging. In order to guarantee the effective recovery of the high-resolution image of the experiment the authors analyze the influence of laser coherence in different modes and the surface properties of diverse materials for diffused targets. For the analysis, the authors choose the single-mode fiber laser as the illumination source and the metal material with high diffused reflectivity as the experimental target to ensure the validity of the experimental results. Based on the above, the authors emulate the camera scanning with a single camera attached to an X-Y translation stage and the experimental system with a working distance of 3310mm is used as an example to image the five-cent coin. The authors also perform speckle analysis for rough targets and calculated the average speckle size using normalized autocorrelation function in different positions. The way of calculating the average speckle size for everyday objects provides the premise for subsequent researches on the image quality evaluation; meanwhile, the coherence of light field and the targets with high-reflectivity under this experiment provide the application direction for the further development of the technique, such as computer vision, surveillance and remote sensing 
